# NGS-Based Genetic Analysis in a Cohort of Italian Patients with Suspected Inherited Myopathies and/or HyperCKemia

**DOI:** 10.3390/genes14071393

**Published:** 2023-07-02

**Authors:** Federica Invernizzi, Rossella Izzo, Isabel Colangelo, Andrea Legati, Nadia Zanetti, Barbara Garavaglia, Eleonora Lamantea, Lorenzo Peverelli, Anna Ardissone, Isabella Moroni, Lorenzo Maggi, Silvia Bonanno, Laura Fiori, Daniele Velardo, Francesca Magri, Giacomo P. Comi, Dario Ronchi, Daniele Ghezzi, Costanza Lamperti

**Affiliations:** 1Medical Genetics and Neurogenetics Unit, Fondazione IRCCS Istituto Neurologico Carlo Besta, 20126 Milan, Italy; federica.invernizzi@istituto-besta.it (F.I.); rossella.izzo@istituto-besta.it (R.I.); andrea.legati@istituto-besta.it (A.L.); nadia.zanetti@istituto-besta.it (N.Z.); barbara.garavaglia@istituto-besta.it (B.G.); eleonora.lamantea@istituto-besta.it (E.L.); lorenzo.peverelli84@gmail.com (L.P.); costanza.lamperti@istituto-besta.it (C.L.); 2Child Neurology Unit—Department of Pediatric Neuroscience, Fondazione IRCCS Istituto Neurologico Carlo Besta, 20133 Milan, Italy; anna.ardissone@istituto-besta.it (A.A.); isabella.moroni@istituto-besta.it (I.M.); 3Department of Neuroimmunology and Neuromuscular Diseases, Fondazione IRCCS Istituto Neurologico Carlo Besta, 20133 Milan, Italy; lorenzo.maggi@istituto-besta.it (L.M.); silvia.bonanno@istituto-besta.it (S.B.); 4UOS di Malattie Metaboliche e Nutrizione, Ospedale dei Bambini Vittore Buzzi, 20154 Milan, Italy; laurafiori69@gmail.com; 5Neuromuscular and Rare Disease Unit, Fondazione IRCCS Ca’ Granda Ospedale Maggiore Policlinico, 20122 Milan, Italy; daniele.velardo@policlinico.mi.it (D.V.); giacomo.comi@policlinico.mi.it (G.P.C.); 6Neurology Unit, Fondazione IRCCS Ca’ Granda Ospedale Maggiore Policlinico, 20122 Milan, Italy; francesca.magri@policlinico.mi.it (F.M.); dario.ronchi@unimi.it (D.R.); 7Department of Pathophysiology and Transplantation, Dino Ferrari Center, University of Milan, 20122 Milan, Italy; 8Lab of Neurogenetics and Mitochondrial Disorders, Department of Pathophysiology and Transplantation, University of Milan, 20122 Milan, Italy

**Keywords:** hyperCKemia, creatine kinase, rhabdomyolysis, skeletal muscle damage, Next Generation Sequencing (NGS), myoglobinuria

## Abstract

Introduction/Aims HyperCKemia is considered a hallmark of neuromuscular diseases. It can be either isolated or associated with cramps, myalgia, weakness, myoglobinuria, or rhabdomyolysis, suggesting a metabolic myopathy. The aim of this work was to investigate possible genetic causes in order to help diagnose patients with recurrent hyperCKemia or clinical suspicion of inherited metabolic myopathy. Methods A cohort of 139 patients (90 adults and 49 children) was analyzed using a custom panel containing 54 genes associated with hyperCKemia. Results A definite genetic diagnosis was obtained in 15.1% of cases, while candidate variants or variants of uncertain significance were found in a further 39.5%. Similar percentages were obtained in patients with infantile or adult onset, with some different causative genes. *RYR1* was the gene most frequently identified, either with single or compound heterozygous variants, while *ETFDH* variants were the most common cause for recessive cases. In one patient, mRNA analysis allowed identifying a large *LPIN1* deletion missed by DNA sequencing, leading to a certain diagnosis. Conclusion These data confirm the high genetic heterogeneity of hyperCKemia and metabolic myopathies. The reduced diagnostic yield suggests the existence of additional genes associated with this condition but also allows speculation that a significant number of cases presenting with hyperCKemia or muscle symptoms are due to extrinsic, not genetic, factors.

## 1. Introduction

Elevated serum Creatine Kinase (CK) levels are considered the hallmark of neuromuscular disorders [1]. Although this biochemical abnormality is related to many inherited diseases, it can also be associated with several non-genetic conditions, including physical exercise, muscle injury, pregnancy, drugs, malignancies, alcohol and other toxins, infections, hyperthermia, thyroid and parathyroid diseases, and hematopathies [2]. Chronic elevation of serum CK (hyperCKemia) is occasionally encountered in apparently healthy individuals. Rowland et al. (1980) [3] coined the term “idiopathic hyperCKemia” for cases with no clinical and histopathological evidence of neuromuscular disease. The association of a high level of CK with other symptoms, such as cramps or myalgia, less often rhabdomyolysis (RM), increases the likelihood of reaching a final diagnosis of muscle-inherited disease. RM is characterized by fulminant skeletal muscle damage resulting in the release of intracellular muscle components into the bloodstream leading to myoglobinuria, and in severe cases, acute renal failure could be a rare but fatal event [4]. Some genetic diseases can be a cause of RM and pose a diagnostic challenge, considering their marked heterogeneity and rarity [5]. Many disorders characterized by hyperCKemia and RM/myoglobinuria are associated with the depletion of muscle energy stores or interruption of the metabolic processes responsible for regenerating ATP [6]. Structural elements of muscle cells (such as the dystrophin-glycoprotein complex) or the excitation-contraction mechanism of the muscle, if altered, can also lead to hyperCKemia and RM [5]. The skeletal muscle ryanodine receptor (RYR1) is also associated with a broad spectrum of hereditary myopathies. Mutations in *RYR1* are related to susceptibility to malignant hyperthermia and are shown to be the most common cause of exertional RM [7]. The first report of recurrent attacks of myoglobinuria was reported in a Swedish man in 1955, but the genetic cause was clarified only in 2008 by single-nucleotide polymorphism (SNP) microarray genotyping [8,9]. Several additional genes have been associated with these clinical conditions. In most cases, inheritance is autosomal recessive, with only a few conditions inherited in an X-linked, autosomal dominant, or matrilinear fashion [5,8]. In many instances, muscle histological and histochemical studies are not able to drive to a specific genetic diagnosis, and the genes involved are extremely long to analyze by direct sequencing. Only in rare metabolic myopathies (i.e., *CPT2* and *ETFDH* deficiency, mitochondrial myopathies) screening of humoral biomarkers may address genetic analysis.

Recently, DNA sequencing techniques have evolved with the introduction of Next Generation Sequencing (NGS), which allows the analysis of multiple genes simultaneously with reduced costs and time. An extensive genetic investigation is therefore possible in pathologies with high genetic heterogeneity. Several papers have been published showing how targeted NGS can increase the diagnostic score in rare heterogeneous diseases [10], although the diagnostic yield is strictly dependent on selected inclusion/exclusion criteria. However, a large fraction of patients with hyperCKemia or clinical suspicion of inherited metabolic myopathies remains without a genetic diagnosis after extensive NGS screenings [5].

In this study, we analyzed 139 samples received by our laboratory for NGS screening by a panel, including genes known to be associated with hyperCKemia, metabolic myopathies, and rhabdomyolysis.

## 2. Patients and Methods

### 2.1. Patients Cohort

A cohort of 139 patients referred from 2018 to 2022 to the outpatients’ clinic of the Foundations IRCCS Istituto Neurologico “Carlo Besta” and IRCCS Ca’ Granda Ospedale Maggiore Policlinico in Milan was analyzed for this study. The cohort included 90 adults (71% males, mean age 38 years, age range 19–69; 29% females, mean age 40, age range 19–59) and 49 pediatric cases (68% males, mean age 11 years, age range 2–17: 32% females, mean age 13, age range 8–17). All the patients presented with cramps, myalgia, and/or increased level of CK (ranging from 2 to >10-fold higher than the control values) with variable muscle weakness. 15/139 patients presented at least one episode of RM. Electromyography (EMG) was performed on 19 patients. 49/139 of the patients underwent muscle biopsy during the diagnostic process: an increase of lipid storage was noticed in 8, mitochondrial alterations were reported in 4, in 3 patients there was an increase of glycogen, and 8 showed a necrotizing myopathy, and no specific abnormalities or mild myopathic changes were identified in the remaining 26 patients’ biopsies. Details on phenotypes, muscle features, and exam findings for each patient are reported in Appendix A, and the most significant findings from the histopathological studies are present in representative images in Appendix A.

### 2.2. Genetic Studies

After informed consent, a blood sample was taken from all the patients (and available family members), and genomic DNA was extracted from leukocytes. We used 250 ng DNA as a template for the construction of a paired-end library, according to the Illumina DNA Prep with Enrichment protocol (Illumina), and using a custom panel (Agilent SureSelect) of genomic regions corresponding to the transcribed sequences of 54 genes (Table 1). These items have been selected amongst genes associated with inherited diseases possibly presenting with hyperCKemia/RM, following a literature review performed in 2018. Genes associated with muscular dystrophies or muscle channelopathies, which do not manifest as pseudo-metabolic myopathy, were not included. Libraries were sequenced on a MiSeq instrument (Illumina). The sequencing reads were aligned to the NCBI human reference genome (GRCh37/hg19) using the Burrows–Wheeler Aligner (BWA). Single nucleotide variants (SNVs) and small insertions/deletions (INDELs) calling were performed using GATK4.1; Variant Interpreter software (Illumina) was used for variants annotation and filtering. Filtering was carried out by applying a series of steps: low-quality variants were filtered out (Illumina Qscore threshold of 30); variants with a minor allele frequency (MAF) ≥1% in the 1000 Genomes Project (http://www.1000genomes.org, accessed on 9 January 2023) and GnomAD databases (https://gnomad.broadinstitute.org, accessed on 9 January 2023) were discharged. Finally, we focused on predicted missense, frameshift, stop-gain or stop-loss, and splice-site variants. For prioritized variants in the final list, we also used bioinformatics tools for pathogenicity prediction (e.g., Polyphen2, SIFT) and variant classification according to the ACMG criteria (https://franklin.genoox.com, accessed on 12 May 2023). These variants were confirmed by Sanger sequencing in patients and assessed in DNAs from available family members.

For interesting cases, after informed consent, skin biopsies were taken and fibroblast cell lines were cultured following standard procedures. Total RNA was isolated from fibroblasts (RNeasy Mini Kit, Qiagen-Italy, Milan) and then reverse-transcribed to cDNA (GoTaq^®^ 2-Step RT-qPCR System). Sequence analysis was performed using the Big-Dye Terminator Ready Reaction Kit version 1.1 on a 3100xl Genetic Analyzer Automatic Sequencer (Applied Biosystems, Life Technologies Europe-Monza, Italy).

## 3. Results

After NGS analysis of our custom gene panel, we selected rare variants (MAF < 0.01) with a predicted effect on the corresponding transcript or protein. All these variants were classified according to the ACMG criteria into five classes: pathogenic (P), likely pathogenic (LP), uncertain significance (VUS), likely benign (LB), and benign (B) [11]. Most of the identified variants in our study (62%) were VUS, while only 21% reached P or LP tiers (Figure 1A). In our cohort, we find no major differences between the average number of variants found in adult or infantile patients and their classification. High genetic and allelic heterogeneity was present (Figure 1B).

In addition to single variant classification, we considered the genetic contribution to disease in each tested individual. Patients were classified as “solved”, hence with a definitive diagnosis, if they had: (i) ≥1 LP-P variants in genes associated with diseases with autosomal dominant (AD) inheritance; (ii) 1 homozygous LP-P or ≥2 heterozygous VUS-LP-P (including at least 1 LP-P) variants in genes associated with autosomal recessively inherited diseases (even in the absence of proof of compound heterozygosity); (iii) 1 LP-P variant in genes causing X-linked recessive diseases (in males). The patients that were classified as “uncertain” including those with a single VUS in genes linked to an AD trait or possibly interesting variants (either 1 LP-P or 2 VUS) in genes associated with AR inherited diseases. Patients were classified as negative or “unsolved” if no variants or only LB/B variants were found.

In short, the genetic screening revealed that only a minority of patients reached a certain molecular diagnosis (16.8% of adults and 12.2% of children). A large fraction of both infantile and adult patients remained doubtful, with ≈40% of uncertain cases. Unsolved cases accounted for about half of adults and children (Figure 2). In the following paragraphs, we report the main and most interesting findings observed in the different subgroups.

### 3.1. Solved Patients

A summary of the genetic findings of the 21 patients classified as “solved” is reported in Table 2, while their demographic, laboratory, and clinical features are reported in Table 3. Among children, we identified variants in four genes that can be considered the genetic cause in the following six cases (6/49, 12.2%):

*ETFDH:* A homozygous variant affecting a splice acceptor site in *ETFDH* (Electron transfer flavoprotein dehydrogenase, OMIM*231675) in a patient (1/C) who presented lipid accumulation in muscle and heart.

*HADHB:* Two heterozygous *HADHB* variants (Hydroxyacyl-CoA dehydrogenase trifunctional multienzyme complex subunit β, OMIM*143450) in one patient (2/C) who presented with diverse episodes of weakness after exertion, an episode of severe limb-girdle and axial weakness with onset due to fever. Segregation analysis showed that both parents carry only one of the two variants found in the proband, confirming the latter as having a biallelic defect.

*RYR1:* Three infantile cases carried variants in *RYR1* (Skeletal muscle ryanodine receptor, OMIM*180901) that can be considered responsible for the phenotype. A heterozygous missense *RYR1* variant was found in one case (3/C) with myalgia, cramps, and rhabdomyolysis. The variant (initially classified as VUS) was inherited from the symptomatic father, leading to a new classification as LP (ACMG criterion PP1: Cosegregation with disease in multiple affected family members). Another similar case (4/C) had a known pathogenic *RYR1* variant inherited from the symptomatic mother. One subject with myopathy (5/C) was compound heterozygous for two VUS (confirmed by segregation analysis in parents, allowing to reclassify the variants as LP based on criterion PM3), suggesting a possible recessive trait.

*LPIN1*: The last infantile case (6/C) with *LPIN1* variants (Mg^2+^-dependent phosphatidic acid phosphohydrolase, OMIM*605518) is an example of a case classified as “uncertain” by the NGS analysis, but that reached a definite diagnosis after additional laboratory investigation (RNA study): it is described in detail in the next section.

In 15 adult patients (15/90, 16.7%), we found likely causative variants in 7 genes:

*ANO5:* One patient (7/A), presenting with myalgia, cramps, and proximal weakness with areflexia, carried two pathogenic variants in *ANO5* (Miyoshi-3 muscular dystrophy, OMIM*608662).

*CPT2:* wo patients (8/A, 9/A) carried rare variants in *CPT2* (Carnitine Palmitoyl Transferase 2, OMIM*600650), either homozygous or possibly compound heterozygous; both presented with myalgia and exercise intolerance, myoglobinuria, and RM.

*ETFDH:* Homozygous or suspected compound heterozygous *ETFDH* variants were found in six adult patients (10/A, 11/A, 12/A, 13/A, 14/A, and 15/A), 4 out of 6 with muscle biopsy positive for lipid accumulation (muscle biopsy was not available for the other 2). Almost all had persistent elevated CK (5/6), and half (3/6) had altered EMG with neurogenic or myopathic signs. Only one case documented dicarboxylic aciduria.

*PYGM:* One patient (16/A) carried a pathogenic homozygous variant in *PYGM* (muscle isoform of glycogen phosphorylase, OMIM*608455); she showed cardiomyopathy, fatigue and cramps and inflammatory necrotizing features at muscle biopsy.

*RBCK1:* A homozygous variant in *RBCK1* (RANBP2-type and C3HC4-type zinc finger containing 1, OMIM*610924) was found in one patient (17/A). As the other *RBCK1* patients previously described [12], she presented with progressive weakness and cardiomyopathy. Moreover, positive PAS staining was reported in histological examinations on muscle biopsy.

*RYR1:* After *ETFDH*, the second most recurrent genetic cause was *RYR1*: two patients (18/A, 19/A) had a single (LP or P) variant, while two cases (20/A, 21/A) harbored two variants (VUS + LP), with unknown phase. They all had elevated CK, weakness, myalgia, cramps, and episodes of RM. In one case (18/A), family segregation analysis showed that the *RYR1* variant was inherited from the mother, who also showed hyperCKemia.

Details on clinical presentations, muscle, and biochemical findings usually associated with these genes are reported in Appendix A. Except for one *RYR1* case, it was not possible to investigate the segregation of other family members of these patients. However, for cases with two heterozygous variants in genes associated with AR diseases, we considered that they are likely to compound heterozygous for the two variants based on their rarity and fitting phenotype. Further future efforts will be made to collect DNA from family members or additional samples for RNA analyses from the probands to experimentally validate the allelic status.

### 3.2. Patients with Uncertain Diagnosis

Several “uncertain” cases were identified among both adult patients (35/90, 38.9%) and children (20/49, 40.8%). All variants (see Appendix A) were confirmed by Sanger sequencing, but segregation analysis or other studies could not be conducted for all patients. Therefore, a final diagnosis was not possible, and further investigations need to be carried out. We report hereinafter an example of additional studies that can lead to a definite diagnosis in such cases; in particular, for those samples with a single interesting variant in a gene associated with AR inherited diseases, it is possible to hypothesize the presence of a second variant, not detected by NGS of the custom panel.

#### 3.2.1. The *LPIN1* Example: From Uncertain to Solved Case by RNA Analysis

We studied an infantile patient (6/C) with marked hyposthenia, elevated CK level (129,000 UI/L; normal level < 50 UI/L), myoglobinuria, muscle pain, and inauspicious outcome; he was the second child of unaffected parents (Figure 3A). We identified heterozygous deleterious variants in two different genes: c.986A > G (p.Tyr329Cys) in *GBE1* (Glycogen Branching Enzyme 1, NM_000158) and c.328C > T (p.Arg110*) in *LPIN1* (Lipin 1, NM_001349206). The *GBE1* variant was present in a public database (rs80338671) and had been previously reported as pathogenic, associated with a loss of ≈50% of enzymatic activity [13]. The *LPIN1* c.328C > T variant (rs747835893) had an extremely low frequency (0.002% in gnomAD) and was predicted to introduce a premature termination codon. It was classified as LP. Notably, both *GBE1* and *LPIN1*-related diseases present autosomal recessive inheritance. All the exons and intron-exon junctions of the two genes were fully covered by NGS (100% with 30× depth of coverage). In order to identify a possible second variant, we analyzed the corresponding transcripts. Through amplification of the whole transcripts from fibroblasts, we did not identify any alteration in *GBE1* while we detected aberrant *LPIN1* transcripts. Their analysis revealed the presence of *LPIN1* transcripts missing exons 19–20 (Figure 3B). Going back to NGS data on genomic DNA, we observed an aberrant coverage profile of intron 18 upstream of exon 19 (Figure 3C). By specific amplification between introns 18 and 20, we identified a 1763 bp deletion, encompassing the whole exon 19, and defined its breakpoints (chr2:11.958.744-11.960.507) (NM_001349206: c.2550-866_2665-30del) (Figure 3D). Segregation analysis in the DNA from family members showed the deletion in the father and the non-sense change in the mother, while two unaffected siblings did not harbor either of the two variants (Figure 3A). Biallelic mutations in this gene are responsible for recurrent acute myoglobinuria, characterized by muscle pain, weakness, and episodic or recurrent attacks of RM triggered by catabolic stress, possibly leading to renal failure. Muscle biopsy was not available to confirm lipid inclusions, which are usually modest and have been reported to be absent in some *LPIN1* mutant subjects [14].

#### 3.2.2. The *CPT2* Example: AR or AD?

We also included in this group of uncertain diagnoses a patient with three VUS (p.Gln304His, Val446Ile, p.Val507Ile) in *CPT2*, a gene typically showing AR transmission. It was not possible to verify the status of the three variants by segregation analysis because this child was adopted (but NGS revealed that two are definitely in cis). Nevertheless, AD inheritance was reported for *CPT2* since some carriers of single heterozygous variants may become symptomatic during exercise [15].

### 3.3. Unsolved Patients

We considered as “unsolved” all patients where we did not identify any rare variant or we found only single heterozygous variants (LB/B/VUS) in autosomal recessively inherited genes. For genes associated with AD inheritance (or X-linked inheritance in males), the presence of a B or LB variant was not considered enough to make a definite/uncertain diagnosis. Overall, this group includes 63/139 of the patients (40/90 adults, 44.4%; 23/49 children, 46.9%) (Figure 2).

## 4. Discussion

The study of the causes underlying persistent hyperCKemia, with or without myoglobinuria and RM, or suspected metabolic myopathy, has always been tricky because of the large number of different pathways possibly altered in these pathological conditions. Nevertheless, this is one of the most frequent conditions that not only neurologists, but clinicians in general, must address. The first works reported from the 90s described only biochemical/histological analyses on a limited number of enzymes [16]. In this respect, new NGS approaches that allow screening a large number of genes described as associated with isolated hyperCKemia or more complex syndromes can be helpful in shedding light on the complex association between genetic variants and clinical manifestation.

We used a custom gene panel containing 54 known genes associated with hyperCKemia, metabolic myopathies, and rhabdomyolysis. According to the role of the corresponding proteins, it is possible to define five main categories/pathways, which include (A) fatty acid oxidation/lipid metabolism; (B) glycogen metabolism; (C) mitochondrial disorders; (D) muscular dystrophies/congenital myopathies. Additionally, disturbances in intramuscular calcium release and excitation-contraction coupling (E) are linked to *RYR1* impairment (Table 1).

(A) Fatty acids are a source of energy; through β-oxidation and the electron transport chain, ATP is generated inside the mitochondria. These processes require numerous enzymes. In patients with defects in this pathway, symptoms of pain and cramps are triggered by prolonged activity of low to moderate intensity. In the case of lipid disorders, accumulating non-metabolized long-chain fatty acids can also damage the membranes [6]. Most of our solved patients with recessive traits have two VUS/LP/P variants in genes belonging to this group. In particular, we identified two compounds with heterozygous variants in *CPT2*, three possible heterozygous compounds and four homozygous variants in *ETFDH*, and compound heterozygous variants in *HADHB* in one little patient. Group A is the most represented, with 10 out of 21 (48%) solved cases.

(B) Through the metabolism of glycogen stored in the muscle, glucose is generated and then used to produce ATP (glycolytic pathway) [6]. Depending on the dysfunctional enzyme, there can be different clinical manifestations; in infantile subjects, different tissues may be affected (with liver, heart, and/or muscle involvement), while adults show predominantly muscle involvement [17]. RM is often triggered by short bouts of intense exercise. More than 18% of our patients carried variants in one of the genes belonging to this group, but the majority were classified as “uncertain” or “negative” because of the predicted poorly deleterious effect of the variants or their inconsistency with a mode of inheritance of the corresponding diseases. Only one case, with a homozygous pathogenic variant in *PYGM*, received a certain molecular diagnosis.

(C) Mitochondrial diseases are very heterogeneous—they can affect any part of the body at any age. Considering only the muscle symptoms, they typically occur immediately after exercise and/or after prolonged activity. In addition to increased CK, elevated lactate levels or lactate/pyruvate ratio may be found. Several nuclear genes are associated with mitochondrial myopathies or multisystem diseases with muscle involvement and often present hyperCKemia. Furthermore, diverse mutations in the mitochondrial DNA have been associated with myopathy and myoglobinuria. The only nuclear gene responsible for recurrent myoglobinuria in mitochondrial myopathy is *LPIN1.* LPIN1 is a phosphatidic acid phosphohydrolase, and its impairment causes a phospholipid composition defect leading to the accumulation of lysophospholipids in the inner mitochondrial membrane, which act as detergents, thus promoting muscle breakdown [8]. We identified a baby patient in our cohort falling into this group and carrying biallelic variants in *LPIN1* (including a large deletion identified by functional studies on the transcript).

(D) Muscle structure and function are maintained thanks to the action of numerous proteins, which, if altered, can cause structural/functional damage to the muscle. Among these, muscular dystrophies are a heterogeneous group of degenerative, progressive hereditary myopathies. The muscles of these patients are much more susceptible to damage and RM events can occur following exercise. We found two patients, the first homozygous for a P variant in *RBCK1* and the second compound heterozygous for *ANO5* LP variants, belonging to this group. In our case series, 13 patients presented variants in *DMD*, which cannot be considered the definitive cause of their disease because they were single VUS in females or benign variants in males. Only one female (26/C) carries a variant classified as pathogenic, but clinical information excludes that it could be the cause of her disease. Furthermore, in these patients with *DMD* variants (both those considered uncertain and those considered unsolved), whenever possible, we excluded dystrophin abnormalities by protein or immunohistochemical studies.

(E) Finally, a large fraction of our patients (34/139) carried variants in the ryanodine 1 receptor (*RYR1*) gene, and many of them have single VUS. RyR1, located on the sarcoplasmic reticulum, along with dihydropyridine sensitive receptor (DHPR) located on the transverse tubules (T tubules) and the sarco/endoplasmic reticulum Ca^2⁺^ ATPase (SERCA), plays a central role in the excitation-contraction process, or the translation of a nerve electrical impulse into muscle contraction; this is a complicated mechanism involving several ion channels and intramuscular pumps. RyR1 dysfunction-related disorders are the most common and best characterized among diseases affecting this process [5]. Mutations in *RYR1* have been associated with different diseases with both dominant and recessive inheritance: susceptibility to Malignant Hyperthermia, Central Core Disease Minicore, and Centronuclear Myopathy with External Ophthalmoplegia. In our cohort, only seven patients with *RYR1* variants can be considered with a definitive diagnosis. Two adult patients and one child had a single heterozygous variant in *RYR1*, with high scores by pathogenic predictions. Furthermore, the parents harboring the same variant presented increased levels of CK: these findings supported a causative variant and dominant inheritance. Four other patients harbored two heterozygous variants and were classified as possible compound heterozygotes, but we could not analyze the relatives because their DNA were not available. In total, 17 subjects with single or two heterozygous VUS were classified as uncertain. *RYR1* is undoubtedly the most frequently identified gene in our analysis, but its high clinical heterogeneity, together with its variable mode of inheritance (either AD or AR), hamper an immediate assessment of the identified variants. In addition, *RYR1* is expressed only in muscle, and transcript analysis cannot be performed on other easily accessible biological samples. The difficulties in identifying a consistent genotype-phenotype correlation in *RYR1* patients have been recently demonstrated by Dosi et al. [18], who tried to cluster with a high-level data integration system a large cohort (75 patients) of individuals with *RYR1* positive molecular diagnosis. Hence, further investigation, particularly segregation in family members, will be required for a diagnostic conclusion.

In our study, a certain genetic diagnosis was obtained in 15.1% of the patients (21/139), confirming the data of some recently published works. However, the diagnostic rate may be higher in selected and specific cohorts. Using an NGS custom panel of genes associated with limb-girdle muscular dystrophies, RM, and metabolic myopathies, in 66 patients, the authors obtained a diagnostic rate of 50%, and VUS were found at 26%, emphasizing the importance of the NGS diagnostic approach in genetically heterogeneous conditions, especially for isolated hyperCKemia [19]. In another cohort of 21 unrelated families with RM, variants in candidate genes were found in 43% of patients by whole exome sequencing (WES). Nevertheless, it was performed on subjects with a high degree of consanguinity (Jewish and Arabic origin). Disease-causing mutations in 8 different genes were found. Among these genes, five were on our panel (*CPT2*, *PFKM*, *PGAM2*, *CACNA1S*, *RYR1*), while three were not (*MYH3*, *SCN4A,* and *AHCY*). *MYH3* and *AHCY* mutations are associated with complex phenotypes, often involving dysmorphic features, mental retardation, and neurologic impairment, which do not overlap with our cohort. *SCN4A* mutations can cause muscle diseases such as congenital myotonia, myasthenic syndrome, and hypokalemic periodic paralysis. HyperCKemia is rarely reported but can be present [20]. This study confirmed the potential of WES to increase the detection rate for monogenic aetiologies and revealed a broad genetic heterogeneity for RM [21]. Finally, an oligogenic model of inheritance was suggested, at least for exertional RM. Genetic results from the WES study on seven adult male subjects with recurrent exertional RM revealed multiple heterozygous variants in genes associated with monogenic metabolic and/or mitochondrial disorders [22].

For a critical review of our results, particularly concerning the low diagnostic score, a series of considerations can be made. Firstly, more extensive laboratory work (e.g., larger gene panels and analysis of mitochondrial DNA-mtDNA) could lead to an increased positive diagnosis. In fact, the major limitation of a gene panel is that it may not include all relevant genes for a specific clinical condition and often requires frequent updates. However, to have uniform data, we analyzed the same 54 genes panel for all the individuals. NGS of the whole mtDNA is feasible, but because of polyplasmy/heteroplasmy, it requires a dedicated analysis, and the diagnostic score is highest when using DNA from an affected tissue (typically muscle) [23]. Secondly, available biological samples influence the diagnostic yield. Being able to deepen the analysis of segregation (e.g., for identifying de novo variants in an autosomal dominant form) or to carry out functional studies (e.g., by transcript analysis to identify the second variant in subjects with only one pathogenic variant in a recessive condition as in the case of *LPIN1*), a diagnostic conclusion could also be possible for additional subjects in our cohort. Lastly, as demonstrated by other works, a WES analysis may be required to identify genetic defects in genes not investigated by our targeted approach. Furthermore, WES could discover new causative genes, or WES data can be reanalyzed later, considering the identification of new disease-causing genes. Likewise, dedicated analyses would be required for genetic defects not detected by NGS, such as the CCTG expansion in myotonic dystrophy type II, reported in a recent diagnostic workflow for patients with minimally symptomatic hyperCKemia [24].

Nevertheless, a substantial proportion of patients presenting with hyperCKemia or with a clinical suspicion of inherited metabolic myopathies remains genetically unsolved, even after WES, suggesting additional pathogenic mechanisms not yet discovered for this condition [5]. HyperCKemia may reflect a combination of a genetic predisposition and environmental triggers, and the presence of an identifiable trigger does not necessarily exclude an underlying genetic cause, further complicating the picture since these two components can be concausative [25]. An interesting hypothesis considers the presence of “risk” polymorphisms that cause an additive effect in the pathogenesis of hyperCKemia or RM; however, studies on more numerous samples should be carried out.

Finally, the diagnostic yield in the genetic analysis of highly heterogeneous diseases is strictly dependent on the inclusion/exclusion criteria of the analyzed subjects. In our study, we included all the patients with increased CK levels or suspected metabolic myopathy and a clinical spectrum ranging from virtually asymptomatic hyperCKemia to cramps and myalgia, up to severe RM. This broad phenotype contributes to explaining the limited percentage of solved cases in our whole cohort. Furthermore, the set of laboratory examinations (muscle biopsy, humoral biomarkers) performed in our cohort was variable. Indeed, the diagnostic yield greatly increased in selected subgroups by taking into account specific features such as positive histological findings. For instance, an altered muscle biopsy was present in 50% of patients with a definite diagnosis, while in indefinite cases, the percentage of patients with biopsy alterations was much lower (equal to 27.8% in uncertain cases and 23.8% in unsolved cases). In particular, increased lipid content was observed in all the patients harboring variants in *ETFDH* for whom a muscle biopsy was available, strengthening this association. Similarly, the percentage of solved cases among subjects presenting RM or myoglobinuria reached 45% (Appendix A).

Despite genetic screening leading to a limited number of solved cases in this group of conditions, it remains of considerable importance. Indeed, some of these patients are at risk for episodes of RM and possibly subsequent risk for acute renal failure; moreover, the identification of the molecular defect can allow genetic counseling, adequate prophylaxis, and treatment for these patients and their family members [21].

## 5. Conclusions

In the present work, a cohort of patients with hyperCKemia, RM, or suspected metabolic myopathies was analyzed by an NGS panel of genes associated with the phenotype. About 15% of patients received a definite diagnosis, but the diagnostic yield was higher in specific subgroups characterized by muscle histological alterations or specific clinical/biochemical symptoms like myoglobinuria and RM. Further analyses with broader approaches, such as WES, need to be performed to exclude a genetic basis for unsolved cases, and functional studies are instrumental in defining the role of the many VUS identified by NGS. However, our and previous studies clearly indicate that a large fraction of subjects with increased CK levels, with or without RM, does not present a genetic (at least a monogenic) disorder.

## Figures and Tables

**Figure 1 genes-14-01393-f001:**
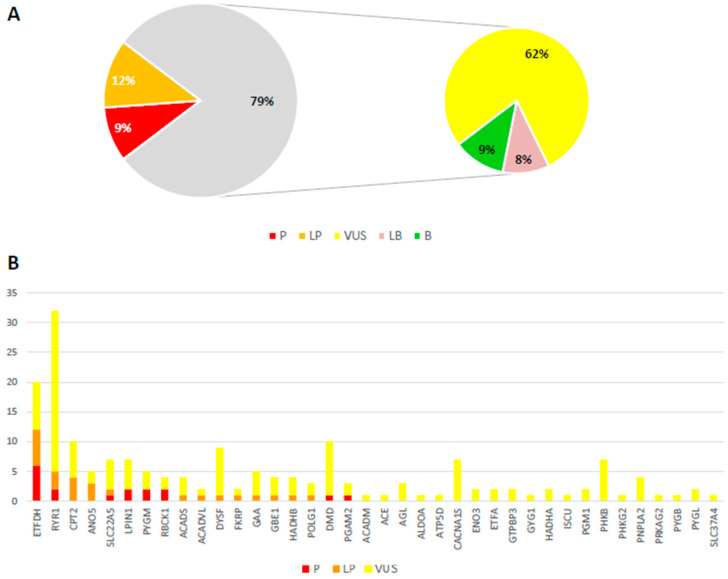
**Type of candidate variants and affected genes identified in this study**. (**A**) Distribution of the identified variants according to the tiers defined by the ACMG criteria (P: pathogenic; LP: likely pathogenic; VUS: variant of uncertain significance; LB: likely benign; B: benign). (**B**) Number of variants (P: Pathogenic; LP: Like Pathogenic; VUS: variant of uncertain significance) found in different (38/54) genes analyzed by our NGS panel.

**Figure 2 genes-14-01393-f002:**
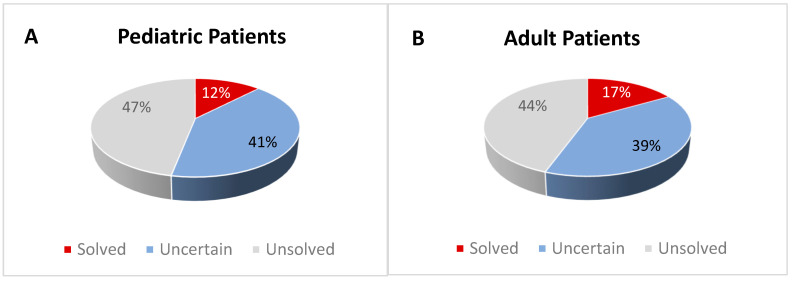
**Distribution of the investigated patients in the selected groups according to their genetic diagnosis.** Patients were classified as “solved” (with a definitive diagnosis), “uncertain” (with an inconclusive diagnosis), or “unsolved” (negative cases). Panel (**A**) reports infantile cases; panel (**B**) reports adult cases.

**Figure 3 genes-14-01393-f003:**
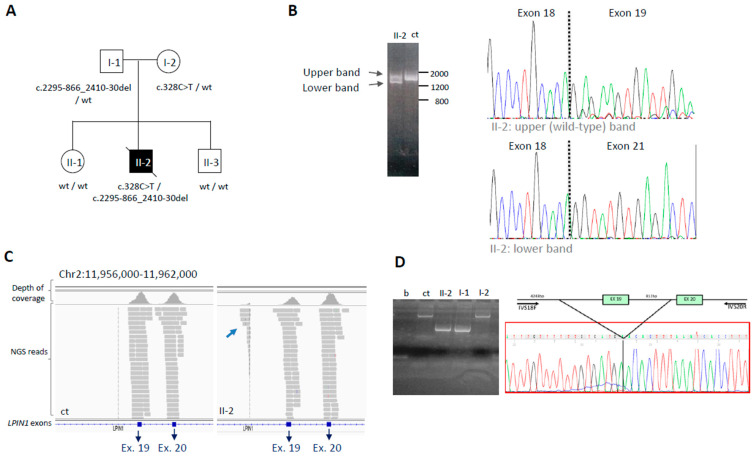
**Deep DNA and RNA analysis of the subject with *LPIN1* variants.** (**A**) Pedigree of the patient 6/C (II-2), carrying two variants in *LPIN1*, and segregation analysis in the family. (**B**) Amplification of the *LPIN1* transcript, which highlights two bands in the proband II-1, while control DNA (ct) has a single band. Sanger electropherograms show that the upper band corresponds to the wild-type transcript, and the lower one corresponds to a transcript missing exons 19–20. (**C**) Snapshots from Integrative Genomic Visualization showing NGS analysis of the genomic region containing *LPIN1* exons 19–20. The lower coverage of exon 19 in II-2 compared to control (ct) is due to a heterozygous exon deletion; in II-2, some reads (arrow) map to the region upstream of the deletion. (**D**) Genomic DNA analysis by PCR using specific primers to define the deletion breakpoints (b = blank sample, ct = control sample, II-2 patient 6/C, I-1: father; I-2: mother). Sanger sequencing of the amplicon containing the deletion reveals it is c.2295–866_2410-30del.

**Table 1 genes-14-01393-t001:** List of 54 genes present in the NGS panel, divided into groups based on their function and associated diseases.

**(A) Disorders of fatty acid oxidation/lipid metabolism**			
**Gene**	**Protein**	**Inheritance**	**Disease**
*ACADM*	Medium-chain acyl-CoA Dehydrogenase	AR	Deficiency of medium chain acyl-CoA dehydrogenase
*ACADS*	Short-chain acyl-CoA Dehydrogenase	AR	Deficiency of short chain acyl-CoA dehydrogenase
*ACADVL*	Very-long-chain acyl-CoA dehydrogenase	AR	Deficiency of very-long-chain acyl-CoA dehydrogenase
*CPT2*	Carnitine palmitoyl-transferase II	AR	Deficiency of Carnitine palmitoyl-transferase 2
*ETFA*	Electron transfer flavoprotein-asubunits	AR	Multiple acyl-coenzyme A dehydrogenase deficiency-Glutaric aciduria type IIA
*ETFB*	Electron transfer flavoprotein-bsubunits	AR	Multiple acyl-coenzyme A dehydrogenase deficiency-Glutaric aciduria type IIB
*ETFDH*	Electron transfer flavoprotein: ubiquinone oxidoreductase	AR	Multiple acyl-coenzyme A dehydrogenase deficiency-Glutaric aciduria IIC
*FLAD1*	Flavin adenine dinucleotide synthetase	AR	Lipid storage myopathy due to flavin adenine dinucleotide synthetase deficiency
*HADHA*	Hydroxyacyl-CoA dehydrogenase/3-ketoacyl-CoA thiolase/enoyl-CoA hydratase, alpha subunit	AR	Mitochondrial trifunctional protein deficiency
*HADHB*	Hydroxyacyl-CoA dehydrogenase/3-ketoacyl-CoA thiolase/enoyl-CoA hydratase, beta subunit	AR	Mitochondrial trifunctional protein deficiency
*PNPLA2*	Adipose Triglyceride lipase	AR	Neutral lipid storage disease
*SLC22A5*	Organic cation transporter 2	AR	Carnitine deficiency, systemic primary
**(B) Disorders of glycogen metabolism**			
**Gene**	**Protein**	**Inheritance**	**Disease**
*AGL*	Amylo-1,6-Glucosidase, 4-Alpha-Glucanotransferase	AR	Glycogen storage disease IIIa/IIIb, glycogen debrancher enzyme
*ALDOA*	Fructose-1,6-bisphosphate aldolase	AR	Glycogen storage disease XII
*ENO3*	Enolase b	AR	Glycogen storage disease XIII
*G6PC*	Glucose-6-phosphatase (G6Pase)	AR	Glycogen storage disease Ia
*GAA*	Acid Alpha-1,4-Glucosidase (Acid maltase)	AR	Glycogen storage disease II
*GBE1*	1,4-Alpha-Glucan Branching Enzyme Amylo-(1,4 to 1,6) Transglucosidase Amylo-(1,4 to 1,6) Transglycosylase	AR	Glycogen storage disease IV, Polyglucosan body disease (adult form)
*GYG1*	Glycogenin-1 (glycosyltransferase)	AR	Glycogen storage disease XV, Polyglucosan body myopathy 2
*GYS1*	Glycogen synthase (mucle)	AR	Glycogen storage disease 0
*GYS2*	Glycogen synthase (liver)	AR	Glycogen storage disease 0
*LDHA*	Lactate dehydrogenase (muscle, A subunit)	AR	Glycogen storage disease XI
*PFKM*	Phosphofructokinase (muscle)	AR	Glycogen storage disease VII, Tarui disease
*PYGB*	Glycogen phosphorylase (brain)	AR	Glucogen storage disease V
*PYGL*	Glycogen phosphorylase (liver)	AR	Glycogen storage disease VI
*PYGM*	Glycogen phosphorylase (muscle)	AR	Glycogen storage disease V, McArdle disease
*PGAM2*	Phosphoglycerate mutase-2 (mucle)	AR	Glycogen storage disease X
*PGK1*	Phosphoglycerate kinase-1	X-linked	Phosphoglycerate kinase 1 deficiency
*PGM1*	Phosphoglucomutase-1	AR	Congenital disorder of glycosylation, type 1t Glycogen storage disease XIV
*PHKA1*	Phosphorylase kinase-a1 (muscle)	X-linked	Glycogen storage disease IXd
*PHKA2*	Phosphorylase kinase-a2 (liver)	X-linked	Glycogen storage disease IXa2
*PHKB*	Phosphorylase kinase-b	AR	Glycogen storage disease IXb
*PHKG2*	Phosphorylase kinase-g (liver, testis)	AR	Glycogen storage disease IXc
*PRKAG2*	Noncatalytic gamma subunit of AMP-activated protein kinase	AD	Glycogen storage disease of heart, lethal congenital
*SLC2A2*	Glucose transporter-like (GLUT2)	AR	Fanconi-Bickel syndrome - Glycogen storage disease XI
*SLC37A4*	Glucose-6-PhosphateTransporter 1	AR	Glycogen storage disease Ib-Ic
**(C) Mitochondrial functions**			
**Gene**	**Protein**	**Inheritance**	**Disease**
*ATP5D*	Mitochondrial ATP synthase F1 complex-delta subunit	AR	Mitochondrial complex V (ATP synthase) deficiency
*DGUOK*	Mitochondrial deoxyguanosine kinase	AR	Mitochondrial DNA depletion syndrome 3 (hepatocerebral type)
*FDX1L*	Ferredoxin 1-like ptotein	AR	Mitochondrial myopathy, episodic, with optic atrophy and reversible leukoencephalopathy
*ISCU*	Iron-sulfur (Fe-S) clusters scaffold protein	AR	Iron-sulphur cluster deficiency myopathy (mitochondrial disorder)
*HSD17B10*	2-methyl-3-hydroxybutyryl Co-A dehydrogenase	X-linked	Neurodegenerative disorder, chorioathetosis with mental retardation and abnormal behavior
*LPIN1*	Phosphatidic acid phosphohydrolase 1	AR	Phosphatidic acid phosphatase deficiency
*POLG*	Polymerase gamma	AR/AD	Mitochondrial DNA depletion syndrome Progressive external ophthalmoplegia
**(D) Muscular dystrophies/congenital myopathies**			
**Gene**	**Protein**	**Inheritance**	**Disease**
*ANO5*	Transmembrane protein 16E Anoctamin 5	AR	Miyoshi muscular dystrophy 3 Muscular dystrophy, limb-girdle, autosomal recessive 12
*CACNA1S*	Calcium channel	AD	Malignant hyperthermia susceptibility 5 Thyrotoxic periodic paralysis Hypokalemic periodic paralysis, type 1
*CAV3*	Caveolin-3	AD	Myopathy, distal, Tateyama type
*CHKB*	Choline kinase	AR	Muscular dystrophy, congenital, megaconial type
*DMD*	Dystrophin	X-linked	Duchenne muscular dystrophy, Becker muscular dystrophy
*DYSF*	Dysferlin	AR	LGMD2B, Miyoshi myopathy
*FKRP*	Fukutin-related protein	AR	LGMD2I
*FKTN*	Fukutin	AR	Fukuyama congenital muscular dystrophy
*RBCK1*	RANBP-Type and C3HC4-Type Zinc Finger-Containing 1	AR	Polyglucosan body myopathy 1 with or without immunodeficiency
*SIL1*	Nucleotide Exchange Factor	AR	Marinesco-Sjogren syndrome
**(E) Disorders of intramuscular calcium release and excitation-contraction coupling**			
**Gene**	**Protein**	**Inheritance**	**Disease**
*RYR1*	Skeletal muscle ryanodine receptor 1	AD/AR	Malignant hyperthermia-susceptibility, Exertional rhabdomyolysis, Congenital myopathy

AD: autosomal dominant; AR: autosomal recessive.

**Table 2 genes-14-01393-t002:** Genetic findings of "solved" patients.

N°/Child or Adult	Mutated Gene/Inheritance	RefSeq Match	Reference Group Genes *	cDNA	Protein	ACMG/Franklin Classification	Zygosity	Familiarity
1/C	*ETFDH*/AR	NM_004453.4	A	c.176-2A>T	splice acceptor variant	P	Homozygous	Not investigated
2/C	*HADHB*/AR	NM_000183.3	A	c.1280G>A c.1370C>T	p.Gly427Glu p.Ala457Val	LP VUS	Compound heterozygous	Mother p.Gly427Glu Father p.Ala457Val
3/C	*RYR1*/AD-AR	NM_000540.3	E	c.10010G>A	p.Arg3337Gln	LP	Heterozygous	Father (s) p.Arg3337Gln Mother negative
4/C	*RYR1*/AD-AR	NM_000540.3	E	c.14918C>T	p.Pro4973Leu	P	Heterozygous	Mother (s) p.Pro4973Leu
5/C	*RYR1*/AD-AR	NM_000540.3	E	c.13490C>G c.4759G>C	p.Pro4497Arg p.Ala1587Pro	VUS VUS	Compound heterozygous	Mother p.Pro4497Arg Father p.Ala1587Pro
6/C	*LPIN1*/AR	NM_001261428	C	c.328C>T c.2395-866_2410-30del	p.Arg110 * p.Glu766_Ser838del	P P	Compound heterozygous	Mother p.Arg110 * Father p.Glu766_Ser838del
7/A	*ANO5*/AR	NM_213599.3	D	c.902G>T c.2516T>G	p.Gly301Val 21 p.Met839Arg	LP LP	Possibly compound heterozygous	Not investigated
8/A	*CPT2*/AR-AD	NM_000098.3	A	c.338C>T	p.Ser113Leu	LP	Homozygous	Not investigated
9/A	*CPT2*/AR-AD	NM_000098.3	A	c.338C>T c.887G>T	p.Ser113Leu p.Arg296Leu	LP VUS	Possibly compound heterozygous	Not investigated
10/A	*ETFDH*/AR	NM_004453.4	A	c.1531G>A c.1832G>A	p.Asp511Asn p.Gly611Glu	VUS LP	Possibly compound heterozygous	Not investigated
11/A	*ETFDH*/AR	NM_004453.4	A	c.250G>A	p.Ala84Thr	P	Homozygous	Not investigated
12/A	*ETFDH*/AR	NM_004453.4	A	c.1249C>T c.1531G>A	p.Gln417Ter p.Asp511Asn	P P	Possibly compound heterozygous	Not investigated
13/A	*ETFDH*/AR	NM_004453.4	A	c.74dupA c.256C>T	p.Tyr25 * p.Arg86Cys	LP VUS	Possibly compound heterozygous	Not investigated
14/A	*ETFDH*/AR	NM_004453.4	A	c.250G>A	p.Ala84Thr	LP	Homozygous	Not investigated
15/A	*ETFDH*/AR	NM_004453.4	A	c.1531G>A	p.Asp511Asn	LP	Homozygous	Not investigated
16/A	*PYGM*/AR	NM_005609.4	B	c.1A>G	p.?	P	Homozygous	Not investigated
17/A	*RBCK1*/AR	NM_031229.4	D	c.896_899delAGTG	p.Glu299Valfs * 46	P	Homozygous	Not investigated
18/A	*RYR1*/AD-AR	NM_000540.3	E	c.14545G>A	p.Val4849Ile	LP	Heterozygous	Father negative Mother (s) p.Val4849Ile
19/A	*RYR1*/AD-AR	NM_000540.3	E	c.11708G>A	p.Arg3903Gln	P	Heterozygous	Not investigated
20/A	*RYR1*/AD-AR	NM_000540.3	E	c.8594T>C c.6226_6228delA AG	p.Val2865Ala p.Lys2076del	VUS LP	Possibly compound heterozygous	Not investigated
21/A	*RYR1*/AD-AR	NM_000540.3	E	c.12700G>A c.4910C>T	p.Val4234Met p.Ala1637Val	LP VUS	Possibly compound heterozygous	Not investigated

* Gene classification according to the groups reported in Table 1: (A) fatty acid oxidation/lipid metabolism; (B) glycogen metabolism; (C) mitochondrial disorders; (D) muscular dystrophies/congenital myopathies; (E) disturbances in intramuscular calcium release. AD: autosomal dominant; AR: autosomal recessive; (s): symptomatic. ACMG classification by Franklin tool [May 2023]: P: Pathogenic; LP: Like Pathogenic; VUS: variant of uncertain significance.

**Table 3 genes-14-01393-t003:** Clinical and laboratory findings and muscle biopsy features of "solved" patients.

N°/ Child or Adult	Age/Sex	Alive	CK	RM/Myoglobinuria	CNS Involvement	Muscle Findings	Other
1/C	13y/M	No	Elevated	No/No	No	Lipid accumulation (muscle and heart)	EMG: myopathy
2/C	11y/F	Yes	Mild increase	No/No	No	Neurogenic	Previous episodes of weakness after exertion. An episode of severe limb-girdle and axial weakness with onset due to fever
3/C	16y/M	Yes	No	Yes/No	No	Not done	Myalgia, cramps
4/C	16y/M	Yes	Mild increase	No/No	No	Absence of alterations	Myalgia and cramps after excercise, family history of hyperckemia and myalgias
5/C	12y/F	Yes	Elevated	Yes/No	No	Not done	Myopathy
6/C	3y/M	No	Elevated	No/Yes	No	Not done	MR: symmetrical muscular inflammation (legs)
7/A	28y/M	Yes	Elevated	No/No	No	Not done	Myalgia, cramps. Proximal Hypostenia (No osteotendinous reflexes)
8/A	25y/M	Yes	Elevated	Yes/Yes	No	Not done	Myalgia and fatigue
9/A	51y/M	Yes	Elevated	Yes/Yes	No	Not done	Myalgia, muscle weakness
10/A	21y/F	Yes	Elevated	No/No	No	Vacuolar myopathy, lipid accumulation	Dicarboxylic aciduria, fatigue and myalgia
11/A	42y/M	Yes	Elevated	No/No	No	Lipid accumulation, altered mitochondria	EMG: myopathy
12/A	29y/F	Yes	Mild increase	No/No	No	Lipid accumulation	EMG: neurogenic signs
13/A	77y/M	Yes	Mild increase	No/No	No	Lipid accumulation	EMG: neurogenic signs
14/A	22y/M	Yes	No	No/Yes	No	Not done	Hypostenia, hypotonia (increased liver enzymes, steatosis)
15/A	unreported/M	Yes	Elevated	No/No	No	Not done	Myalgia, cramps
16/A	43y/M	Yes	Elevated	No/Yes	No	Inflammatory necrotizing	Fatigue and cramps
17/A	27y/F	Yes	Mild increase	No/No	No	PAS+	Cardiomyopathy and muscle weakness
18/A	25y/M	Yes	Elevated	Yes/No	No	Not done	Cramps, muscle weakness
19/A	40y/M	Yes	Elevated	Yes/No	No	Not done	Myalgia, muscle weakness
20/A	48y/M	Yes	Elevated	Yes/No	No	Not done	EMG: myopathy
21/A	24y/M	Yes	Mild increase	No/No	No	Myogenic signs. Normal electron microscopy	Myalgia, dyspnoea, muscle pain. Psychomotor delay

RM: rhabdomyolysis; CK: hyperCKemia; CNS: central nervous system; A: adult subjects: C: children; M: male; F: female; PAS+: positive Periodic acid–Schiff staining; EMG: Electromyography; MR: muscle magnetic resonance.

## Data Availability

Data supporting the reported results (e.g., vcf file of the targeted NGS) are available from the corresponding author upon reasonable request.

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
