# Peer review of "NGS-Based Genetic Analysis in a Cohort of Italian Patients with Suspected Inherited Myopathies and/or HyperCKemia"

_genes, 2023, doi:10.3390/genes14071393_

Round 1
Reviewer 1 Report
In this manuscript Invernizzi et al. perform a NGS-based genetic analysis in a cohort of hyperCKemia patients with the aim to investigate the possible genetic causes of this disease.
There are some major points that would require some improvements:
· In the abstract, introduction/aims sections, it would be advisable to include the current state of these studies. What is the current diagnostic percentage for this type of pathology?
- In the abstract, results sections, RYR1 is mentioned as the most frequently identified gene, but it is not clarified whether it is dominant, recessive, or considering both types of inheritance. It would be advisable to clarify this.
- In the abstract, conclusion section, it is mentioned that there are cases with hyperCKemia or muscle symptoms due to non-genetic factors. It would make more sense to include this observation and further explanation in the discussion section rather than in the abstract. Without additional context provided in the abstract, it may appear contradictory to the paper's intention.
- In the Materials and Methods section, specifically in the subsection on patient cohort, the authors state that the study includes 90 adult patients and 49 pediatric patients. It would be necessary to specify the age ranges for both groups, as well as the gender.
- The authors mention different histopathological observations. They do not refer to membrane abnormalities in any of the cases. Muscular dystrophies could explain an increase in CK levels. Were patients with membrane protein abnormalities not included in the cohort? Were these abnormalities not visualized/studied?
- In the Methods section, it would be advisable to reserve a subsection or at least a paragraph under "Genetic studies" regarding the construction of the panel. How were the 54 genes selected? What criteria were used to select these genes over others?
- It would be advisable to include a figure depicting a flowchart of the steps followed from patient inclusion in the cohort to the final result (genetic diagnosis).
- In the Methods section, under Genetic studies, it is mentioned that "bioinformatic tools" were used to evaluate variants. Were they used for all variants or only for VUS? What specific tools were used? The authors should provide more details on this point.
- In the Methods section, under Genetic studies, it is explained that RNA was extracted from fibroblasts. Although muscle biopsy is mentioned earlier, there is no specific mention of a skin biopsy being performed. Both should be included, specifying that informed consent was obtained.
- In the the bar graph of figure 1, P should appear below LP.
- In Table 2, the gene names should be italicized.
- In the Results section, under Solved patients, each paragraph for each detailed case comment could start with the gene name. For example, "ETFDH: A homozygous [...]" and "HADHB: Two heterozygous [...].", giving more importance to the mutated genes. The authors should provide justification for the order in which the cases are presented (alphabetical, number of variants, or classification).
- In the specific case of the three pediatric patients with RYR1 mutations, it should be specified based on what evidence according to the ACMG guidelines the variant classification changed from VUS to LP.
- All OMIM codes should include an asterisk if referring to the gene *231675 instead of OMIM_231575.
- In line 238, it is explained that the variant introduces a premature termination codon. How was the impact of this variant evaluated? What bioinformatic tools were used?
- In Table 2, it would be advisable to include the date of the latest classification. Since the evidence upon which the variants are classified can vary, it is recommended to reflect the most recent consultation date.
- A figure including the most significant findings from the histopathological studies should be included.
- In the last paragraph of section 3.1, it is mentioned that despite not being able to conduct familial studies, cases with two heterozygous variants in genes associated with recessive inheritance were considered compound heterozygotes. Without functional/experimental studies, this conclusion is a significant assumption. The authors should include, at the very least, a proposal for further studies.
- In line 228, it is stated "to hypothesize the presence of a second variant, not detected by NGS." This statement is incorrect, and I understand that the authors are referring to a second variant not identified by their panel. NGS would include whole-genome sequencing studies, which could identify variants not studied in their panel.
- In line 254, the term "aberrant coverage profile" contradicts what is explained in line 240, which states "fully covered." The first statement (line 240) could be replaced with the coverage percentage or modified to avoid contradicting both sentences.
- Although it is specified in lines 296-297 that the panel did not include genes related to muscular dystrophies or channelopathies, this should be explained earlier. It is recommended to include this clarification in the section or subsection that explains how the panel was constructed.
- In line 329, it is stated that the LPIN1 patient has "two biallelic variants." I believe this is an error: it should either be "two variants" or "biallelic variants."
- It is advised to be more specific in line 394 regarding "laboratory work." Providing more details about the specific laboratory techniques or procedures would be beneficial.
- The discussion does not include a reflection on the possibility that some cases in the cohort may be explained by mutations in new genes not yet associated with this type of symptomatology. It is recommended to include a reflection on this aspect. It would also be advisable to include a reflection on the possible modification or update of the panel used.
- In the Conclusions section, it is mentioned that future approaches would involve performing WES for unsolved cases. It could also be mentioned that functional studies, especially for variants of uncertain significance (VUS), would aid in achieving a molecular diagnosis.
There are some minor points that require some attention:
- Introduction, line 50: An "and" is missing before "hematopathies."
- It would be advisable to include a reference in the sentence on lines 71-72: "In most of the cases, [...]" or "matrilinear fashion."
- It would be more appropriate to refer to the "length" instead of the "size" of the gene. "Long" instead of "big" in line 74.
- Why is "8/A, 9/A" underlined in line 194?
- Línea 186: Mg2+ se debería sustituir por Mg2+
- In line 254, it is recommended to include a reference that supports the statement "have been reported to be absent in some LPIN1 mutant subjects."
- In line 396, "de novo" should be italicized.
Author Response
>We thank the Reviewers for their overall positive comments and suggestions, which have prompted us to improve the quality and clarity of the paper.
We submitted the new version also with all “tracked changes” (as non-published material) so the editors/reviewers can easily see which amendments have been made.
REV#1
In this manuscript Invernizzi et al. perform a NGS-based genetic analysis in a cohort of hyperCKemia patients with the aim to investigate the possible genetic causes of this disease.
 
There are some major points that would require some improvements:
- In the abstract, introduction/aims sections, it would be advisable to include the current state of these studies. What is the current diagnostic percentage for this type of pathology?
>Because this is a complex issue, depending on inclusion/exclusion criteria and type of genetic screening, the part related to diagnostic yield was inserted and debated in the discussion section (pages 13-14). We added a short sentence in the introduction, but it is not possible to precisely estimate a diagnostic percentage for this clinical condition.
- In the abstract, results sections, RYR1 is mentioned as the most frequently identified gene, but it is not clarified whether it is dominant, recessive, or considering both types of inheritance. It would be advisable to clarify this.
>We added that this sentence refers to both single or compound heterozygous variants. It corresponds to findings reported in Figure 1B.
- In the abstract, conclusion section, it is mentioned that there are cases with hyperCKemia or muscle symptoms due to non-genetic factors. It would make more sense to include this observation and further explanation in the discussion section rather than in the abstract. Without additional context provided in the abstract, it may appear contradictory to the paper's intention.
>This information was added in order to “justify” the reduced diagnostic yield, which was not unexpected but rather regarded as likely. Indeed, we mentioned “non-genetic” factors right away in the introduction (line 3).
- In the Materials and Methods section, specifically in the subsection on patient cohort, the authors state that the study includes 90 adult patients and 49 pediatric patients. It would be necessary to specify the age ranges for both groups, as well as the gender.
>We added information about age mean and ranges, as well as gender distribution, in both adult patients and pediatric patients.
- The authors mention different histopathological observations. They do not refer to membrane abnormalities in any of the cases. Muscular dystrophies could explain an increase in CK levels. Were patients with membrane protein abnormalities not included in the cohort? Were these abnormalities not visualized/studied?
> First, I would like to underline that muscle biopsy was obtained from a minority (only about 1/3) of patients in our cohort. However, immunostaining was done in available muscle biopsies and patients with evident membrane protein abnormalities at immunostaining observations were not included in this study but investigated by targeted gene screening. Immunoblotting was not performed routinely hence mild/partial membrane abnormalities could be not detected.
- In the Methods section, it would be advisable to reserve a subsection or at least a paragraph under "Genetic studies" regarding the construction of the panel. How were the 54 genes selected? What criteria were used to select these genes over others?
>Genes were selected in 2018 based on review of the scientific literature, focusing on the disease groups reported in table 1 (fatty acid oxidation/lipid disorders; glycogenosis; mitochondrial disorders; muscular dystrophies/congenital myopathies). In order to have uniform data, for all the individuals we analysed the same 54 genes panel (not considering updated lists).
Comparing our panel with the lists currently reported in PANELAPP (https://panelapp.genomicsengland.co.uk), 74% of our genes are present in the panelapp “Rhabdomyolysis and metabolic muscle disorders” and 79% in the panelapp “Glycogen storage disease”.
- It would be advisable to include a figure depicting a flowchart of the steps followed from patient inclusion in the cohort to the final result (genetic diagnosis).
>The inclusion criteria for this study were quite broad, reflecting the real cohort of patients sent to our lab for this NGS screening. Because of the wide clinical spectrum and different laboratory examinations of the patients, it is hard to define a flowchart. Furthermore, based on economic and practical aspects (it is cost-effective and less time-consuming if applied to a large number of subjects) the use of a gene panel was considered as a first-tier genetic test for all the cohort without considering alternative genetic analysis.
- In the Methods section, under Genetic studies, it is mentioned that "bioinformatic tools" were used to evaluate variants. Were they used for all variants or only for VUS? What specific tools were used? The authors should provide more details on this point.
>We added more information on this matter.
- In the Methods section, under Genetic studies, it is explained that RNA was extracted from fibroblasts. Although muscle biopsy is mentioned earlier, there is no specific mention of a skin biopsy being performed. Both should be included, specifying that informed consent was obtained.
>We added a mention to the skin biopsy, and the informed consents.
- In the the bar graph of figure 1, P should appear below LP.
>Figure 1 was modified according to reviewer’s suggestion.
- In Table 2, the gene names should be italicized.
>Done.
- In the Results section, under Solved patients, each paragraph for each detailed case comment could start with the gene name. For example, "ETFDH: A homozygous [...]" and "HADHB: Two heterozygous [...].", giving more importance to the mutated genes. The authors should provide justification for the order in which the cases are presented (alphabetical, number of variants, or classification).
>We added the gene name at the beginning of each case. The genes are reported in alphabetical order, keeping children and adult separate. Subject 6/C (LPIN1) is an exception, because initially classified as “uncertain” but bacame a “solved” case after functional studies.
- In the specific case of the three pediatric patients with RYR1 mutations, it should be specified based on what evidence according to the ACMG guidelines the variant classification changed from VUS to LP.
>Based on the ACMG criterion PP1 (Cosegregation with disease in multiple affected family members) the case 3/C was reclassified. The variant in case 4/C was a known Pathogenic variant; the case 5/C had two variants and we considered the criterion PM3 (For recessive disorders, detected in trans) after segregation analysis.
- All OMIM codes should include an asterisk if referring to the gene *231675 instead of OMIM_231575.
>Done
- In line 238, it is explained that the variant introduces a premature termination codon. How was the impact of this variant evaluated? What bioinformatic tools were used?
>Based on the NM_001349206 transcript, the variant c.328C>T introduces a stop codon p.Arg110*. It was predicted by Variant Interpreter, the software we used for variants annotation (see Methods) but the same is expected for any bioinformatic tool.
- In Table 2, it would be advisable to include the date of the latest classification. Since the evidence upon which the variants are classified can vary, it is recommended to reflect the most recent consultation date.
>Done
- A figure including the most significant findings from the histopathological studies should be included.
>Representative images from histological studies are reported in supplementary figure S1.
- In the last paragraph of section 3.1, it is mentioned that despite not being able to conduct familial studies, cases with two heterozygous variants in genes associated with recessive inheritance were considered compound heterozygotes. Without functional/experimental studies, this conclusion is a significant assumption. The authors should include, at the very least, a proposal for further studies.
>We agree with the reviewer. We added a sentence about this point.
- In line 228, it is stated "to hypothesize the presence of a second variant, not detected by NGS." This statement is incorrect, and I understand that the authors are referring to a second variant not identified by their panel. NGS would include whole-genome sequencing studies, which could identify variants not studied in their panel.
>The reviewer is right. We modified the sentence.
- In line 254, the term "aberrant coverage profile" contradicts what is explained in line 240, which states "fully covered." The first statement (line 240) could be replaced with the coverage percentage or modified to avoid contradicting both sentences.
>We added coverage percentage (100% with 30x depth of coverage), which however refers to exons and intron-exon junctions. We now specified that we observed aberrant coverage profile of intron 18, upstream of the exon 19.
- Although it is specified in lines 296-297 that the panel did not include genes related to muscular dystrophies or channelopathies, this should be explained earlier. It is recommended to include this clarification in the section or subsection that explains how the panel was constructed.
>The sentence was added in paragraph 2.2 and removed from discussion.
- In line 329, it is stated that the LPIN1 patient has "two biallelic variants." I believe this is an error: it should either be "two variants" or "biallelic variants."
>The reviewer is right. Thanks for noticing the error.
- It is advised to be more specific in line 394 regarding "laboratory work." Providing more details about the specific laboratory techniques or procedures would be beneficial.
>In the following lines, a series of additional laboratory analyses are reported: functional studies, RNA analysis, WES, laboratory examinations (muscle biopsy, humoral biomarkers). To avoid repetitions, in this very first sentence, we added added only a mention to larger gene panels and analysis of mitochondrial DNA (as possible alternative “first tier” approaches).
- The discussion does not include a reflection on the possibility that some cases in the cohort may be explained by mutations in new genes not yet associated with this type of symptomatology. It is recommended to include a reflection on this aspect. It would also be advisable to include a reflection on the possible modification or update of the panel used.
>Thank you for the suggestion. We added few considerations on this matter.
- In the Conclusions section, it is mentioned that future approaches would involve performing WES for unsolved cases. It could also be mentioned that functional studies, especially for variants of uncertain significance (VUS), would aid in achieving a molecular diagnosis.
>Added.
 
There are some minor points that require some attention:
- Introduction, line 50: An "and" is missing before "hematopathies."
- It would be advisable to include a reference in the sentence on lines 71-72: "In most of the cases, [...]" or "matrilinear fashion."
- It would be more appropriate to refer to the "length" instead of the "size" of the gene. "Long" instead of "big" in line 74.
- Why is "8/A, 9/A" underlined in line 194? It was a typo
- Línea 186: Mg2+ se debería sustituir por Mg2+
- In line 254, it is recommended to include a reference that supports the statement "have been reported to be absent in some LPIN1 mutant subjects."
- In line 396, "de novo" should be italicized.
>All minor points have been corrected.
Reviewer 2 Report
This is an interesting and original manuscript presented by Invernizzi et al. The authors presented a high quality and well-written manuscript with original content with potential key clues for clinical practice. Other groups have performed similar studies previously, however data from different populations are important for clinical practice. Some points should be evaluated another time by the authors at this time:
1. I suggest authors to try to make it clear in their manuscript's title the group which is studied in this manuscript. When presenting hyperCKemia in the title, it could give a false idea that the study will present cases of isolated hyperCKemia, which is not the case. The authors included a large and heterogenous group of patients at different age ranges and individuals with mild to moderate elevation in serum creatine kinase levels. My suggestion for the title in this case could be: "NGS-based genetic analysis in a cohort of Italian patients with suspected inherited myopathies".
2. I suggest authors to include a table (supplemental content, for example) describing the association of genes with clinical presentation (rhabdomyolysis, isolated hyperCKemia, progressive myopathy phenotypes...).
3. Another interesting point for discussion would be to try to correlate some features observed in the sample. Are genetic findings more commonly observed in cases with any type of clinical manifestation? Or in cases without clinical features (isolated hyperCKemia)? Or in patients with CNS involvement? Or in cases with abnormal muscle biopsy studies? This type of information can bring key elements for the clinician during practice and diagnostic work-up of child and adult patients.
Author Response
>We thank the Reviewers for their overall positive comments and suggestions, which have prompted us to improve the quality and clarity of the paper.
We submitted the new version also with all “tracked changes” (as non-published material) so the editors/reviewers can easily see which amendments have been made.
REV#2
This is an interesting and original manuscript presented by Invernizzi et al. The authors presented a high quality and well-written manuscript with original content with potential key clues for clinical practice. Other groups have performed similar studies previously, however data from different populations are important for clinical practice.
Some points should be evaluated another time by the authors at this time:
 
- I suggest authors to try to make it clear in their manuscript's title the group which is studied in this manuscript. When presenting hyperCKemia in the title, it could give a false idea that the study will present cases of isolated hyperCKemia, which is not the case. The authors included a large and heterogenous group of patients at different age ranges and individuals with mild to moderate elevation in serum creatine kinase levels. My suggestion for the title in this case could be: "NGS-based genetic analysis in a cohort of Italian patients with suspected inherited myopathies".
> We used tolerant inclusion criteria, which reflect the real cohort of patients sent to our lab for this NGS screening. While most of the patients have suspected inherited myopathies, some of them presented only increased levels of CK. Hence, we think it is more appropriate a title containing “suspected inherited myopathies and/or hyperCKemia”
- I suggest authors to include a table (supplemental content, for example) describing the association of genes with clinical presentation (rhabdomyolysis, isolated hyperCKemia, progressive myopathy phenotypes...).
>According to the reviewer’s suggestion, we added a supplementary table (Tab. S2) reporting a summary of main clinical presentations, muscle and biochemical findings associated with the genes identified in the “solved” group based on information present on the OMIM catalog.
- Another interesting point for discussion would be to try to correlate some features observed in the sample. Are genetic findings more commonly observed in cases with any type of clinical manifestation? Or in cases without clinical features (isolated hyperCKemia)? Or in patients with CNS involvement? Or in cases with abnormal muscle biopsy studies? This type of information can bring key elements for the clinician during practice and diagnostic work-up of child and adult patients.  
>Supplementary Table 3 (Tab. S2 in the previous version) reports such comparisons, with percentages of solved cases in subgroups with positive findings at muscle biopsy examination; rhabdomyolysis/myoglobinuria; or hyperCKemia. For instance, the diagnostic yield was high in patients with rhabdomyolysis or myoglobinuria. Almost all the patients did not present with CNS involvement.
Round 2
Reviewer 2 Report
The authors have addressed all the changes and suggestions brought previously during the review process and it markedly improved the manuscript in several ways. I have no additional comments at this point.